# Two-dimensional Simulation of Motion of Red Blood Cells with Deterministic Lateral Displacement Devices

**DOI:** 10.3390/mi10060393

**Published:** 2019-06-12

**Authors:** Yanying Jiao, Yongqing He, Feng Jiao

**Affiliations:** School of Chemical Engineering, Kunming University of Science and Technology, Kunming 650500, China; 18388195203@163.com (Y.J.); jiaofeng0526@163.com (F.J.)

**Keywords:** red blood cells, deterministic lateral displacement, trajectories, row shift

## Abstract

Deterministic lateral displacement (DLD) technology has great potential for the separation, enrichment, and sorting of red blood cells (RBCs). This paper presents a numerical simulation of the motion of RBCs using DLD devices with different pillar shapes and gap configurations. We studied the effect of the pillar shape, row shift, and pillar diameter on the performance of RBC separation. The numerical results show that the RBCs enter “displacement mode” under conditions of low row-shift (∆*λ* < 1.4 µm) and “zigzag mode” with large row shift (∆*λ* > 1.5 µm). RBCs can pass the pillar array when the size of the pillar (*d* > 6 µm) is larger than the cell size. We show that these conclusions can be helpful for the design of a reliable DLD microfluidic device for the separation of RBCs.

## 1. Introduction

Human blood consists of plasma and mainly red and white blood cells and platelets [1]. Through continuous circulation, blood provides oxygen and nutrients, removes metabolic waste from tissue, regulates body pH and temperature, in addition to other biological functions. RBCs are the most critical component, and have a direct impact on hemodynamics and hemorheology. RBCs are highly deformable due to their biconcave, seedless, and highly flexible membrane [2]. Researchers has recognized the deformability of RBCs as an inherent indicator of diseases such as diabetes and malaria. The separation of deformable RBCs through Lab-on-a-chip techniques, which is based on the formers’ mechanical properties, is becoming an essential process for medical research and clinical disease diagnosis [3,4,5]. Further understanding of the dynamic behavior of the RBCs in a microchannel, and exploring new methods by which to efficiently separate RBCs from plasma, are urgent to the development of biomedical engineering.

In medical research such as liquid biopsies and cytopathology, researchers have begun to focus on the direct isolation of target cells from whole blood. In 2018, a new method for continuously focusing and separating biological particles directly with shear-induced diffusion from whole blood was successfully demonstrated; the method effectively combines the inherent complexity of blood with the migration from the flow inertia while causing little pollution [6]. They also demonstrated a new multi-flow micro-fluidic (MFM)system for unlabeled circulating tumor cells (CTCs) from the peripheral blood of patients, and proposed a promising alternative method by which to realize CTC capture. Their results show that the method will offer the possibility of achieving chemical individuality treatment [7].

Moreover, several techniques for cell separation have been proposed, such as label-free discrimination and fractionation of cell populations [8], fluorescence-activated cell sorting (FACS), magnetic-activated cell sorting (MACS), and centrifugation separation. Fluorescence-activated cell sorting (FACS) is an active sorting method in which tested cells are stained with a specific fluorescent dye to fluoresce the cells with a complementary fluorophore-conjugated antibody. The detected fluorescence data can be used to characterize the immune situation, size, and cell type, providing rich data support for analyses of gene expression. Magnetically-activated cell sorting (MACS) is a passive separation technique. Magnetic particles are introduced to label cells to bind specific proteins on cells and separate the sample from non-magnetic cells under the action of an external magnetic field [8,9]. Although FACS and MACS can provide rich data for high-throughput screening, these two methods are not widely available due to the limitations of cost of their labeled antibody magnetic nanoparticles and sheath fluids [10]. The macroscopic separation method consists of centrifugation, which uses different concentrations of blood cell components and different centrifugal forces at different settling rates. However, because of its reactivity to environmental changes, centrifugation may alter the immune properties [11]. The phase separation method based on the hematocrit effect in a microchannel with multiple outlets has also been adopted to separate RBCs. Yin et al. [2] investigated the separation process of multiple RBC flows through a symmetric microvascular bifurcation model with different cell deformability, aggregation, and hematocrit. Their symmetric bifurcation model is relatively simple, given the limitations of the 2-D model.

With the development of microfabrication technology, the post arrays of the pillars in a microchannel have been used for the selective delivery on the proteins [12] and drugs [13], and to separate RBCs [3], known as ‘‘deterministic lateral displacement” (DLD) [14,15]. The method has the advantages of requiring lower analytical reagent dosages, shorter detection periods, and yielding higher levels of precision, and this technology opens up the possibility of experimental microchip diagnostics replacing traditional blood tests. By collecting sample fractions to quantify the separation efficiency at each outlet, Xavier et al. [16] found that the purities can reach 88.5% and 98.3% for large and small beads, respectively. Their experimental results demonstrated that the DLD has outstanding potential for high purity separation.

Zhang et al. [4] numerically studied the influences of the pillar shapes on the dynamic behavior of the RBCs in the DLD devices, including cylindrical, diamond, and triangle. Sharp obstacles can significantly enhance the deformability of cells; their sharper edges can cause more significant distortion of RBCs. The cells around triangles bend more strongly than around diamonds, which may be related to DLD-based deformable sorting. Li et al. [17] considered a cylindrical microfluidic bifurcation channel, where three-dimensional fluid dynamics codes were used to simulate the trajectories of RBCs. Since healthy RBCs have the characteristic of high deformability, and the distortions induced by bifurcation are considerable, the healthy ones cannot uniformly distribute in bifurcated microchannels like rigid RBCs. That means that the deformability of RBCs has a significant influence on blood separation, and that healthy RBCs have an extremely high separation efficiency.

Fu et al. [18] combined the lattice-Boltzmann method (LBM), the immersion boundary method (IBM), and the discrete element method (DEM) to calculate the separation of particles of different sizes and shapes (sphere, triangle, diamond, and pentagon) through square and round pillars. Kruger et al. [19] investigated the shapes of RBCs between pillars using the capillary number Ca to indicate the deformability. Ca-dependent trajectories have been observed, and the direct collision of RBCs with the pillars may increase separation. Hou et al. [20] examined a continuous filtration method for the separation of the infected red blood cells (iRBCs) in a microfluidic device. iRBCs behave like white blood cells and move toward the side walls due to cell-cell interactions.

In this paper, we simulate the motions of RBCs in DLD devices by considering their deformability. Firstly, we changed the pillar shapes (circular and triangular) and their arrangements to evaluate the trajectories, surface stresses, and velocities of the rigid RBCs (regarding them as microspheres). We then introduced capillary number Ca to represent the deformability of RBCs, and discussed the variation of Ca under the configuration of the triangular pillars. Our work on the DLD device provides underlying understanding for the efficient separation of the RBCs in the future.

## 2. Simulation Methods and Models

We adopted the COMSOL^®^ Multiphysics software as the computational tool, which is relatively well-suited to the calculation of microspheres manipulation [21]. We optimized the simulation code of [22], in which the separation trajectories in a DLD device are accurately predicted using the finite element method. Furthermore, we studied the velocity field, surface stress, trajectories, and velocity of red blood cell movement. During the simulation, it was shown that when the height of the microspheres/cells is consistent with the thickness of the device, the geometric nature of the microspheres/cells can be simplified to two-dimensions [20]. Also, compared to the 3-D model, the 2-D model can greatly reduce the number of calculations. Additionally, the friction between the microspheres and the device wall can be neglected.

### 2.1. The DLD Model

Figure 1 shows a typical schematic of a DLD device. The microfluidic channel has one inlet on the left and two outlets on the right; the separated particles/cells will flow out of one of the outlets. To enhance the analysis, we chose two different pillar shapes (circle and the triangle) as the calculation cases. The detailed dimensions of the pillar posts are shown in Figure 1.

Table 1 lists the channel length, width, the pillar diameter, the height, and the obstacle spacing of the DLD device. Table 2 lists the physical properties of the fluid and the RBCs. Two working fluids were tested.

#### 2.1.1. Governing Equations

In this two-dimensional simulation, we need to calculate an incompressible liquid flow by solving the Navier-Stokes equations [23,24] and the continuity equation:
(1)$$\rho \frac{\partial u_{fluid}}{\partial t}=\nabla \left[-pI+\mu \left(\nabla u_{fluid}+{\left(\nabla u_{fluid}\right)}^{T}\right)\right]-12\frac{\mu u_{fluid}}{d_{{}_{z}}^{2}}+F$$
(2)$$\rho \nabla \times u_{fluid}=0$$
where $\rho \frac{\partial u_{fluid}}{\partial t}$ represents the unsteady inertia force (N/m^3^), $12\frac{\mu u_{fluid}}{d_{z}^{2}}$ represents the nonlinear inertia force (N/m^3^), and ***F*** represents the volume force. *ρ* represents the fluid density (kg/m^3^), *p* is the pressure (Pa), ▽() represents the gradient operator, ***I*** is the unit diagonal matrix, *μ_f_* is the dynamic fluid viscosity (Pa·s), and *d_z_* represents the channel height (mm). When gravity or other volumetric forces are not considered, ***F*** = 0. ***u_fluid_*** represents the fluid velocity field (m/s).

In a microfluidic system, the flow rate is small and the Reynolds number (*R_e_* << 100) is expressed as
(3)$$R_{e}=\frac{lU\rho }{\mu },$$
where *l* is the characteristic length of the rectangular channel (*l* = 109 μm), and *U* is the average velocity.

#### 2.1.2. Boundary Conditions for Fluid-solid Interaction (FSI)

In a fluid-solid coupling boundary setting, the solid boundary is affected by the viscous force and flow pressure. Therefore, the velocity of the fluid can be used to derive the displacement rate of change of the solid microspheres.
(4)$$u_{fluid}=u_{w}$$
(5)$$u_{w}=\frac{\partial u_{solid}}{\partial t}$$
(6)σ⋅n=Γ⋅n
(7)$$\Gamma =\left[-pI+\mu \left(\nabla u_{fluid}+{\left(\nabla u_{fluid}\right)}^{T}\right)\right]$$
where Γ is the force on the solid boundary, which includes the fluid pressure and viscous resistance, and ***n*** denotes the outward normal bound. At the fluid-solid coupling boundary, the fluid velocity, ***u_fluid_***, is equal to the rate of change of the solid displacement, ***u_w_***. ***u_solid_*** represents the solid displacement field (m/s), ***u_w_*** represents the displacement rate of change of the solid microspheres and satisfies the no-slip condition at the channel wall, and *σ* is the Cauchy stress.

#### 2.1.3. Initial Conditions

The fluid flow in the channel is fully developed laminar flow, and is driven by the pressure difference. To ensure that the nonlinear solver has the best possible convergence at the beginning, we set the inlet velocity to be non-constant. According to the parabolic characteristics of the laminar flow velocity distribution, we used the following formula to calculate the normal inlet velocity:
(8)$$U=6u_{0}(H-Y)Y/H^{2}$$
where ***u*_0_** is the average flow velocity at the inlet (m/s), *H* is the channel height (mm), and *Y* is the value of the *y*-coordinate of the center of the microsphere/RBCs.

At the exit, a fixed boundary condition, also known as the Dirichlet boundary condition, was used to determine the value of the pressure.
(9)$$\left[-pI+\mu \left(\nabla u_{fluid}+{\left(\nabla u_{fluid}\right)}^{T}\right)\right]n=-\overset{\wedge }{p}n$$

### 2.2. Red Blood Cell Model

Due to their non-expandable nature, RBCs can preserve their area and arc length in two dimensions. The deformability of cells depends on the elasticity of the cell membrane, the viscosity of the cytoplasm, and the applied flow rate [3,25]. Therefore, for the simulation of the cell screening process, it was necessary to consider the deformability of RBCs. Young’s modulus, which is the modulus of elasticity detected by atomic force microscopy, is a physical quantity used to characterize the surface properties of cells in the biological field. It is determined only by the physical properties of the material itself; the larger Young’s modulus, the greater the rigidity of the cells, and the more difficult it is to deform them.

#### 2.2.1. Linear Elastic Material

If the rigid microspheres underwent small levels of deformation and were subjected to a low load, their displacement and deformation satisfy the linear elastic momentum conservation (see the Equation (11)) and the governing equilibrium equation [23,24]. In this simulation, the Young’s modulus *E* = (0.25 ± 0.08) kPa, the Poisson’s ratio *v* = 0.3 for RBCs.
(10)$$\rho \frac{\partial u_{solid}^{2}}{\partial t^{2}}-\nabla \cdot \sigma =F_{\upsilon }$$
(11)$$\sigma =J^{-1}FSF^{T}$$
(12)$$F=\left(I+\nabla u_{solid}\right)$$
(13)J=det(F)
(14)$$\varepsilon =\frac{1}{2}\left[{\left(\nabla u_{solid}\right)}^{T}+\nabla u_{solid}+{\left(\nabla u_{solid}\right)}^{T}\nabla u_{solid}\right]$$
(15)$$S-S_{0}=C:\left(\varepsilon -\varepsilon_{0}-\varepsilon_{inet}\right)$$

Equation (10) represents Newton’s equation of motion; Equation (11) is the linear elastic stress-strain law, and Equation (14) represents the strain-displacement (compatibility) equation. ***F****_v_* is the boundary force and *J* as the stiffness matrix. *ε* is the infinitesimal strain tensor (we also use *ε* to present the row shift fraction in the discussion section); finally, *C* is the stiffness matrix.

#### 2.2.2. The Capillary Number

To consider the factors that affect cell deformation, such as cell membrane, cytoplasm, and the flow rate of the surrounding fluid, we introduced a dimensionless number, capillary number *Ca*, by combining the characteristics of these aspects [3].

The bending stiffness *k_b_* of a healthy RBC is 1 × 10^−19^ and the effective radius *R_eff_* is 2.5 μm [1,3]. The maximum flow velocity *U*_max_ between the pillars is from 10 μm/s to 10 mm/s, and the post-gap *G* = 1 μm. For healthy RBCs, the capillary number *Ca* range is [0.0375, 375], while the *Ca* range of pathological RBCs will increase nearly 10 times due to their stiffness, which means that the value is [0.0038, 37.5] [26]. Therefore, the deformation of RBCs can be judged by the capillary number *Ca*; the larger the capillary number *Ca*, the more easily the RBCs are deformed.

### 2.3. The Mesh and Independence Verification

After creating a mesh of the domain and discretizing the equation, we use the Any Lagrangian-Eulerian (ALE) method to describe the interface between the fluid and the RBCs. The geometry of the domain can be defined by moving the mesh, and the deformation and movement of the boundary are the same. COMSOL can calculate the new grid coordinates of the channel region based on the moving boundary and mesh smoothing of the structure, and solve the Navier-Stokes equations in the moving coordinate system in the mesh and fluid domain. The microsphere meshes in the domain move and deform freely with the simulated motion, and are automatically divided.

Also, we need to conduct grid independence verification to select the appropriate mesh. The most basic method to change the mesh size; we will choose this mesh scale when the difference between the results of two consecutive mesh scales reaches a point at which it can be ignored. Table 3 gives the corresponding degrees of freedom for solving the various equations at different grid scales.

Figure 2a plots the rate of change of displacement in the X direction for circular microspheres with *t* = 0 to *t* = 0.3 seconds at different grid scales. The grid is very close and the error is small in terms of regularity, refinement, ultra-fineness and extreme refinement. To consider the computational efficiency and the solver convergence, the ultra-thinning ratio is used for subsequent simulations. According to the COMSOL mesh quality detection method, the meshing is consistent with the rule that the mesh quality is close to 1, meaning that the mesh quality is good.

Figure 2b is a meshing diagram and shows enlarges the microsphere area. The mesh is divided into free triangle meshes, and the ultra-fine meshing method is selected. Inside the obstacle, the moving mesh will change with the deformation of the obstacle; in the outer boundary of the watershed, the deformation in all directions is set to zero. Therefore, the initial mesh at *t* = 0 s is not evenly divided, but it can be seen that the mesh is evenly distributed around the microspheres. The mesh is looser on the fluid domain, while the fluid-solid boundary has a denser and smaller cell grid.

## 3. Results and Discussion

To simplify the simulation, we consider the RBCs to be microscopically deformed microspheres by ignoring their deformability in order to study the deflection effect of DLD devices on different sized microspheres. Three kinds of microspheres with different diameters were used for numerical simulations; the trajectories of the microspheres flowing through the DLD microchannel were analyzed.

### 3.1. The Trajectories of Microspheres

In a DLD device, the microspheres will separate according to the arrangement of the pillars under the laminar flow at low Reynolds number.

From the Figure 3, there are two main modes in which the rigid microspheres move between the gaps of pillar arrays. It is interesting to note that the position of the pillars will change the trajectories of the microspheres, leading their trajectories into a “zigzag mode”; however, the trajectories return to the original state after three rows of obstacles. When the radius is greater than the width of the channel, the microspheres adopt “displacement mode”, and may not be able to enter channel 1 of the fluid in the gap, thus behaving differently in the pillar array [27].

Compared with the longitudinal rectangular microchannels described in reference [27], we used horizontal microchannels in the numerical simulation to analyze the trajectories of the microspheres. We studied the factors affecting the separation of microspheres in the DLD device, the row shifts fraction *ε*, and the diameter of the microspheres *D*. The row-shift fraction *ε* is expressed as
(16)ε=Δλλ,

As shown in Figure 4, ∆*λ* is row shift and *λ* is the center-to-center distance.

There is a critical diameter *D_c_* between the “displacement mode” and the “zigzag mode”. Based on the gap size and microsphere diameters, the critical diameter of the microspheres is [28],
(17)$$D_{c}=1.4G\varepsilon^{0.48},$$
where *G* is the post-gap, indicating the distance between the two pillars.

The result shows the occurrence of “displacement mode” at low row shifts (∆*λ* ≤ 2.5 µm) and “zigzag mode” for larger row shifts (∆*λ* ≥ 3 µm). This means that the separation trajectories of the microspheres is dependent on the row shifts in the presently-studied DLD device.

To investigate the effect of the complex pillar array on the separation of microspheres, we used different arrangements of pillars, i.e., by varying the row shift, the post-gap and the center-to-center distance between the two pillars. In Figure 4a, the flow pattern of microspheres 1.4 μm and 3 μm in diameter demonstrates “displacement mode” when ∆*λ* = 1.0 µm, *D_c_* = 1.95 μm, *G* = 4 μm, and *ε* = 0.1111. In Figure 4b, the trajectories of microspheres with the same diameter adopt “zigzag mode” when ∆*λ* = 4.0 µm, *D_c_* = 3.79 μm, *G*=4 μm, and ε = 0.4444. This indicates that the trajectories of the microspheres are related to the row shift, which is consistent with the conclusions found in the literature.

Compared to [20], we found that the diameter of the microspheres affects their trajectories; we also selected different diameters of the microspheres in our simulation to verify this result. Three different diameters of microspheres were released from the same position in order to analyze the trajectories when staggered pillars were used. From top to bottom, the microspheres have diameters of 1.0 μm, 0.4 μm and 0.2 μm with the ∆*λ* = 1.5 µm, *D_c_* = 1.88 μm, *G* = 3 μm, and *ε* = 0.1875 respectively, and the trajectories of the three microspheres is in “zigzag mode”. When ∆*λ* = 3.5 µm, *D_c_* = 2.82 μm, *G* = 3 μm, and *ε* = 0.4375, it was found that microspheres with diameters of 0.1 μm, 0.2 μm, 0.4 μm, 1 μm and 2 μm have different displacement rates, but that the trajectories are the same in “zigzag mode”. Therefore, in this arrangement of pillars in Figure 5, the diameter of the microspheres does not change their trajectories; therefore, we suspect that this indicates a new law between “zigzag mode” and “displacement mode” regarding row shifts in these pillar arrangements.

To explore the critical row shifts between the “displacement mode” and the “zigzag mode” in this pillar arrangement, the same diameters, i.e., 1.0 μm, 0.4 μm and 0.2 μm, of the microspheres are in ∆*λ* = 1.0 µm, *D_c_* = 1,55 μm, *G* = 3 μm, and *ε* = 0.125 and another in ∆*λ* = 1.4, *D_c_* = 1.82 μm, *G* = 3 μm, and *ε* = 0.175 again in Figure 6. The results show that they all lead to “displacement mode”, and that in the arrangement of the pillars, the “displacement mode” shifts at the low row shifts (∆*λ* ≤ 1.4 µm), and the “zigzag mode” shifts at the larger row shifts (∆*λ* ≥ 1.5 µm). With the trajectories of microspheres changing from “zigzag mode” to “displacement mode”, the critical diameters become smaller, and the row shift fraction becomes larger.

The microspheres showed periodic variations in this DLD device. We mainly studied microsphere separation in “zigzag mode” using staggered pillars. In this section, we considered RBCs as being rigid, and simulated the microspheres with the same parameters as those of the RBCs. We believe that this arrangement can effectively predict the trajectory of RBCs during separation. We analyzed the velocity and surface stress of different microsphere shapes (circular, elliptical, triangular and diamond) and the effect of the shape of the pillars.

### 3.2. The Surface Stress on the Microsphere

The deformability of RBCs is mainly due to stress on the surface. Next, we will discuss the stress induced by the flow. RBCs can be damaged or even rupture under complex loading forces, including viscous resistance and the pressure of the fluid. Therefore, it is necessary to analyze the surface stress on the microspheres in order to determine whether the microspheres will remain intact in DLD devices with a pillar structure [29].

The Young’s modulus causes elastic deformation by affecting the level of stress on the microspheres. The surface stress of the microspheres is scalar, calculated from a stress tensor, and can be used to determine whether the microspheres remain intact when subjected to the loading force. In Figure 7a, when *t* = 0 s, at 0.126 s, 0.156 s, 0.171 s, and 0.19 s, the position of the microspheres of different shapes is shown in different colors. The maximum and minimum stress points are also presented (Figure 7b,c). When the microspheres are far from the pillars in the DLD device, the stress distribution is almost constant and uniform. The surface stress increases with increasing pressure. Circular and triangular microspheres are always symmetrical when they are in movement, so the maximum surface and minimum surface stress values are close to each other. Because of their different long and short axes, and the fact that microspheres collide with the pillars in the DLD device, the surface stress of elliptical and rhomboid microspheres will change abruptly. Under the low elasticity of the microspheres, surface stress is insufficient to cause apparent deformation.

### 3.3. The Velocity of the Microspheres

According to the five moments shown in Figure 7 above, we found that the velocity of the microspheres is only related to the viscous resistance at the entrance region. Since the viscous resistance is negligible, we consider the initial velocity of the microspheres to be close to the inlet velocity of the fluids (***u*_0_** = 12 μm/s).

Next, we observed the velocity of the liquid in the DLD device. High velocities were measured in the gaps between the pillars. From the velocity of the microspheres, it was observed that the initial velocity of the microspheres was close to the inlet flow velocity, and in the case of a low level of velocity resistance, the shear stress of the fluid on the microspheres was negligible. This confirms that the microspheres have excellent follow-up performance in DLD devices. The result shows that the pillar structure can increase the velocity of the microspheres.

From Figure 8, we can see when the microspheres pass theough the DLD device, their velocity increases, with the maximum velocity occuring in the gaps between the pillar posts. When flowing between the pillars, the velocity was lower; this is caused by the immense surface stress required to produce negative resistance. When the microspheres approach the outlet, the velocity will gradually converge, eventually becoming zero.

### 3.4. The Effect of Pillar Shapes

We investigated the effect of circular and triangular pillar structures on the flow characteristics in DLD devices. Figure 9 shows that there are some zero-velocity areas between the two pillars in these two shapes of pillar posts. We found that the fluid velocity in the circular pillar posts was much greater than that of the triangular ones. We thus optimize the shape of the pillars and analyzed the effect of velocity on the motion and stress levels of the microspheres.

The paraments in these DLD devices are all in ∆*λ* = 3.5 μm, *D_c_* = 2.82 μm, *G* = 3 μm, and *ε* = 0.4375 in Figure 10. The liquid gap between the microspheres and the pillars are clear, while the boundary is slightly weaker. We can see that the microspheres tend to move closer to the pillars in the triangular DLD device, and that the stress level is uniform. This indicates that triangular pillars can increase the levels of interaction with the microspheres, significantly affecting their trajectories.

According to the setting of Young’s modulus and Poisson’s ratio, it was found from the simulation that the deformation of the microspheres was almost consistent with that of RBCs. To better observe RBC deformation in the DLD device with triangular pillars, we decided to combine the RBC model with the triangular pillars to attain the trajectory of the RBCs in the DLD device. And the Figure 11 shows the schematic diagram of the RBCs in 2-D and 3-D.

### 3.5. The Deformation of Red Blood Cells

We analyzed the deformability of RBCs; asphericity needs to be introduced her eto describe the deviation from the spheric geometry [28],
(18)$$\delta =\frac{{\left(\lambda_{1}-\lambda_{2}\right)}^{2}}{{\left(\lambda_{1}+\lambda_{2}\right)}^{2}},$$
where *λ*_1_ and *λ*_2_ are the square roots of two non-zero eigenvalues of the radius of the rotation tensor. The *δ* from 0 to 1 reflects the extent to which perfect RBCs or microspheres changed in terms of elongation. In the 2-D model, *δ* equalled 0.29.

A DLD device with triangular pillars and with a gap of *s* = 3 μm and height of *D* = 5 μm was selected to analyze the level of deformation of RBCs. It was found that there was no clear flow layer between the RBC model and the pillars, indicating that the DLD device has considerable sensitivity. In Figure 12a, the RBCs stayed near the top of the pillar for a long time, and could not continue to move. To explore the relationship between the trajectories of RBCs and the pillar scale, we compared the size of the pillars with a gap of *s* = 10 μm and a height of *D* = 15 μm.

In Figure 12b, it can see that the RBCs are able to pass through the DLD device of the pillar structure when *D* > 6 μm. Simultaneously, by increasing the diameter and the gap of the pillars, the flow velocity of the pillars decreases from the top to the bottom. The velocity of the RBCs near the pillars is less affected by the flow field, and the deformation is negligible.

To characterize the separation model of RBCs, the function of row shift fraction *ε* and capillary number *Ca* was used to verify the trajectories of the microspheres. It was found that the boundary between the displacement model and the “zigzag model” was very close to a simple exponential function [19].
(19)$$\varepsilon (Ca)=2.9+5.4e^{-1.72C_{a}}$$

According to Equation 20, ε(*Ca*) = 3 μm, *Ca* = 2.1 within the “zigzag mode” and the trajectory of RBCs is “zigzag mode” in the row shift ε = 3 μm. ε(*Ca*) = 4 μm and *Ca* = 0.9 in “displacement mode” and when the trajectory of RBCs is in “displacement mode” in the row shift fraction ε = 4 μm. Following Ref. [19], we know that the softer and more rigid gradient of the RBC yield *Ca* ranges from 0.4 to 1.2.

## 4. Conclusions

We studied RBC separation with a DLD device using the finite element simulation. This research will provide an essential reference for further research of RBC separation using DLD devices in the future. The following conclusions were obtained:(a)The trajectories of the RBCs are related to the row shifts. We found that the RBCs enter “displacement mode” under conditions of low row-shift (∆*λ* < 1.4 µm) and “zigzag mode” with large row-shift (∆*λ* > 1.5 µm).(b)The velocity and the surface stress of the microspheres are related to the shape of the pillars. The triangular pillars will produce high velocities and the uniform stress on the microspheres, which will enhance separation.(c)By considering the deformation of RBCs, the row shift fraction *ε* and the capillary number *Ca* are used to determine the RBC separation mode.

## Figures and Tables

**Figure 1 micromachines-10-00393-f001:**
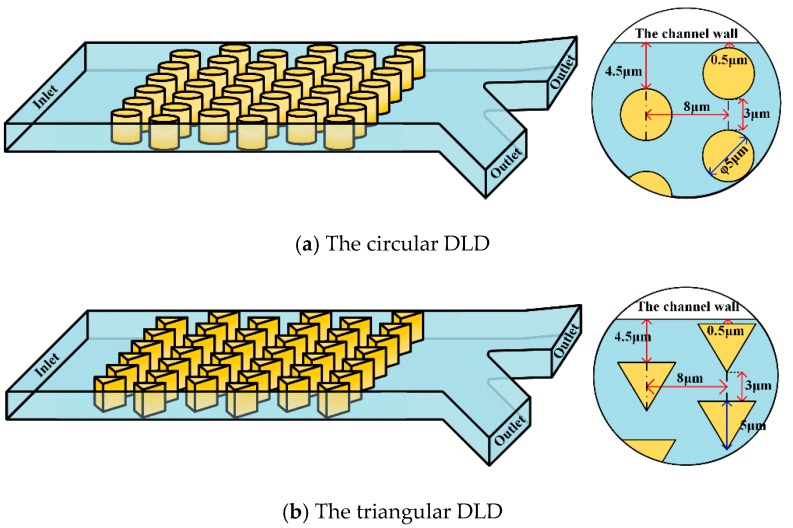
Schematic of a deterministic lateral displacement array device. The microfluidic channel has an inlet on the left and two outlets on the right. The circular and triangular obstacles in the channel are made of polydimethylsiloxane (PDMS). The obstacles are periodically arranged, and each row is horizontal in terms of one row and offset. A liquid solution carrying rigid microspheres of different sizes flows through the passage from the inlet. The pillar shape within the DLD device is (**a**) circular and (**b**) triangular.

**Figure 2 micromachines-10-00393-f002:**
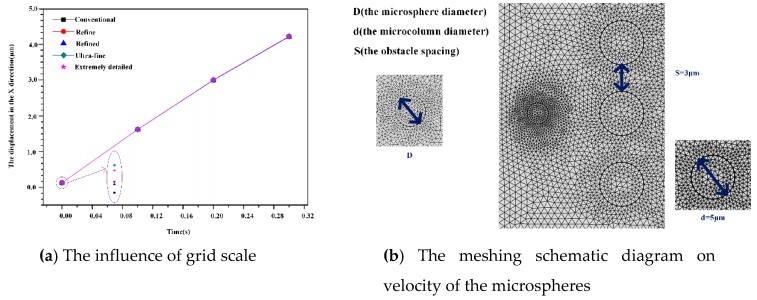
Grid-independent verification (**a**) describes the variation of the velocity of the microspheres along the x-direction at five different grid scales, and (**b**) plots the ultra-fine grids used for the simulation of this number of times.

**Figure 3 micromachines-10-00393-f003:**
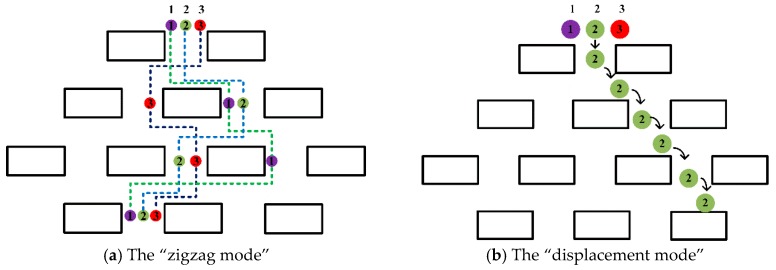
A schematic diagram of a human-defined barrier matrix with three fluid channels. (**a**) The three microspheres in the channel (shown in purple, green, and red, respectively) are incompatible. After three rows of obstacles, the trajectories of the microspheres return to the original channel position, following a zigzag pattern. (**b**) The microspheres with a radius which is greater than the width of the channel follow a displacement pattern, passing through the streamline at the center of the microspheres. The arrow marks the trajectories of the microsphere.

**Figure 4 micromachines-10-00393-f004:**
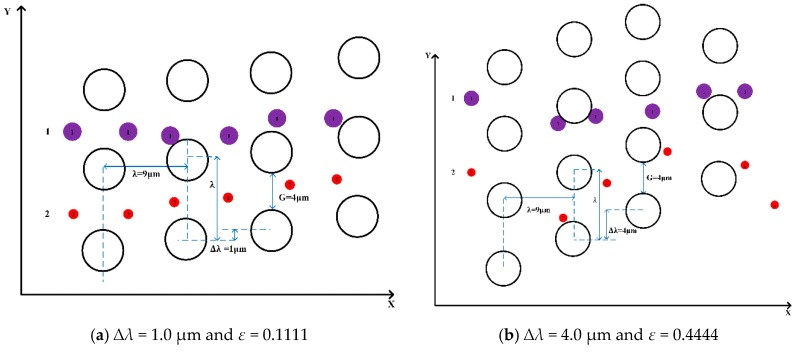
Two types of microspheres with diameters of 1.4 μm and 3.0 μm, marked in red and purple. (**a**) The trajectories of the microspheres are “displacement mode” in a circular pillar; the different row shift fraction is *ε* = 0.1111 and ∆*λ* = 1.0 µm. (**b**) The trajectories of the microspheres are in “zigzag mode” in a circular pillar; the row shift fraction is *ε* = 0.4444 and ∆*λ* = 4.0 µm.

**Figure 5 micromachines-10-00393-f005:**
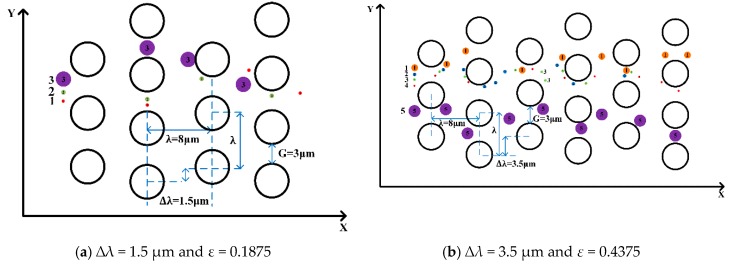
The trajectories of microspheres in a circular pillar are in “zigzag mode”. (**a**) The row shift is ∆*λ* = 1.5 µm and *ε* = 0.1875. We indicate three microspheres in the channel (in purple, green, and red) of different sizes. (**b**) The row shift is ∆*λ* = 3.5 µm and *ε* = 0.4375. Five fluid channels and five differently sized microspheres (shown in orange, green, blue, red, and purple, respectively) are labeled.

**Figure 6 micromachines-10-00393-f006:**
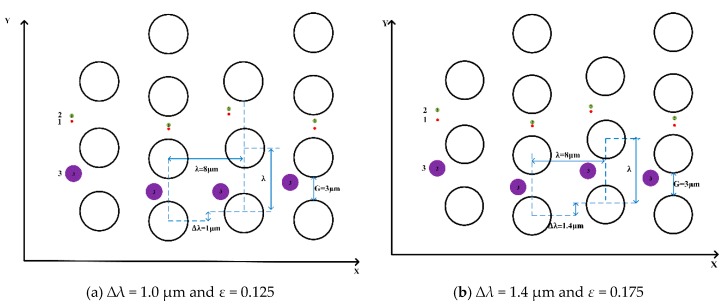
The trajectories of microspheres in a circular pillar array are in “displacement mode”. Marked with two fluid channels, the three microspheres in the channel (shown here in purple and red, respectively) are different in size. (**a**) The row shift is ∆*λ* = 1.0 µm and *ε* = 0.125. (**b**) The row shift is ∆*λ* = 1.4 µm and *ε* = 0.175.

**Figure 7 micromachines-10-00393-f007:**
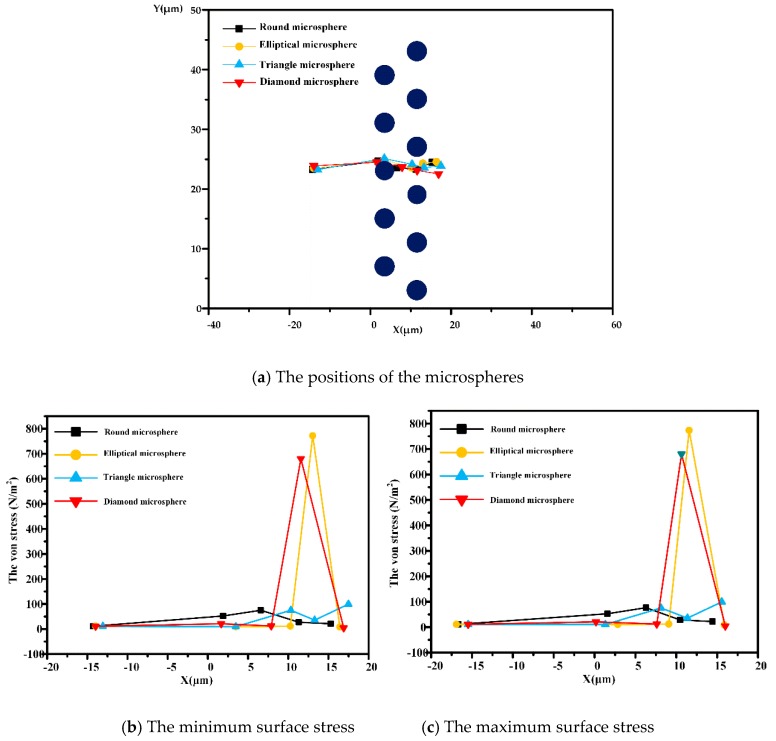
The surface stress on round, elliptical, triangular and rhomboid microspheres (at *t* =0 s, 0126 s, 0.156 s, 0.171 s, and 0.19 s). (a) The microspheres are in different positions at five moments. The minimum and maximum stresses (von stress) are shown in (**b**) and (**c**), respectively.

**Figure 8 micromachines-10-00393-f008:**
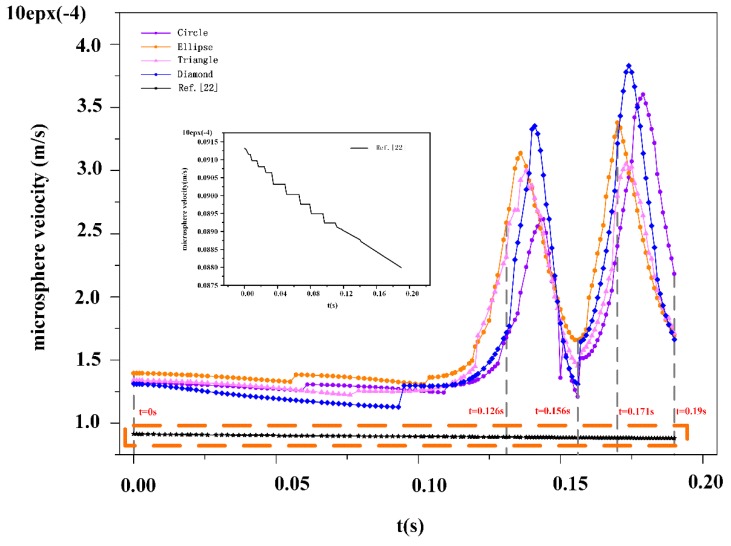
The velocity of the microspheres changes over time. Four kinds of microspheres with circular, elliptical, triangular, and diamond shapes were compared with the data of the reference [22], and a series of time points were selected to analyze their velocity: *t* = 0 s, 0126 s, 0.156 s, 0.171 s, and 0.19 s.

**Figure 9 micromachines-10-00393-f009:**
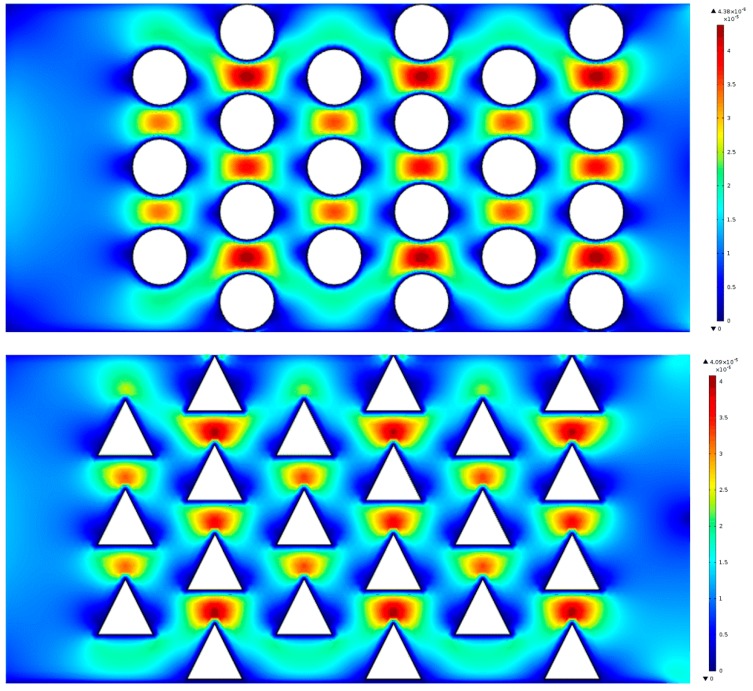
The flow velocity distribution within different shapes of the pillars. The row shift fraction is the same as *ε* = 0.1111. All velocity in the flow direction (x-axis) is represented by the maximum velocity values (m/s) observed in the DLD array, where red indicates a high flow rate region and dark blue a boundary with a no-slip condition, i.e., a zero-flow velocity region.

**Figure 10 micromachines-10-00393-f010:**
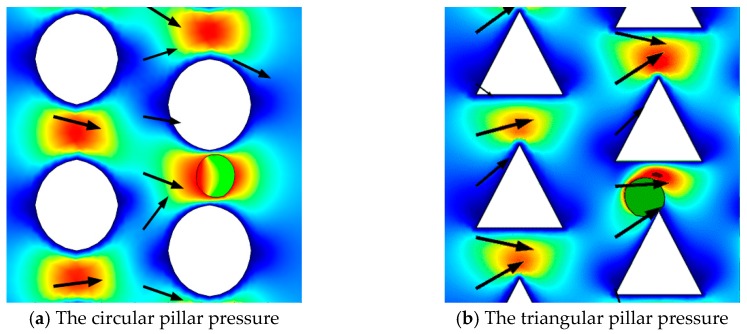
The movement of microspheres identical in size in two types of DLD devices.

**Figure 11 micromachines-10-00393-f011:**
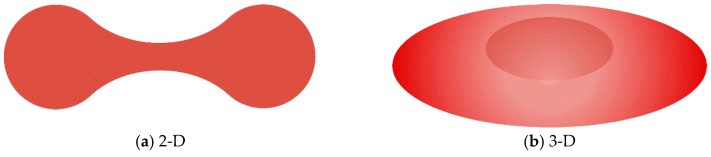
The RBCs model (2-D and 3-D).

**Figure 12 micromachines-10-00393-f012:**
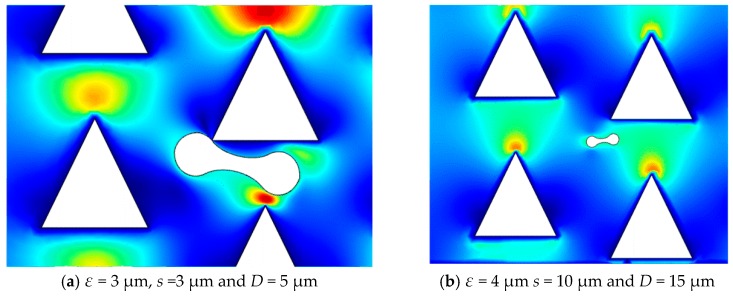
Snapshot of RBCs in the triangular arrays with *s* = 3 μm and *D* = 5 μm and *s* = 10 μm and *D* = 15 μm.

**Table 1 micromachines-10-00393-t001:** The geometric parameters of the DLD array.

The Setting Parameters	The Size (μm)
*L* (the channel length)	109
*W* (the channel width)	54
*d* (the round obstacle diameter)	5
*h* (the equilateral triangle obstacle)	5
*S* (the obstacle spacing)	3

**Table 2 micromachines-10-00393-t002:** The physical properties of the RBCs and working fluids.

The Working Fluid	The Density (kg/m^3^)	The Dynamic Viscosity (Pa·s)	The Young’s Modulus (Pa)	The Poisson’s Ratio
Water	1000	0.001	2.16e9	0.414
RBCs	1090	-	2.5e2	0.3
PDMS	970	0.001	3e9	0.49

**Table 3 micromachines-10-00393-t003:** The grid division scales.

The Cell Type	Maximum Unit Size (μm)	Minimum Unit Size (μm)	Maximum Unit Growth Rate	Curvature Factor	Narrow Area Resolution	Solving the Degree of Freedom
Conventional	2.67	0.119	1.15	0.30	1	11,161
Refine	2.08	0.0594	1.13	0.30	1	11,621
Refined	1.66	0.0238	1.10	0.25	1	16,132
Ultra-fine	0.772	0.00891	1.08	0.25	1	20,140
Extremely detailed	0.398	0.00119	1.05	0.20	1	48,624

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
