# Peer review of "Two-dimensional Simulation of Motion of Red Blood Cells with Deterministic Lateral Displacement Devices"

_micromachines, 2019, doi:10.3390/mi10060393_

Round 1

Reviewer 1 Report

The manuscript reports simulation results of deterministic lateral displacement (DLD) devices. Authors tried to investigate the effects of pillar shape, row shift and pillar diameter and gap on particle moving trajectory. While it is helpful to use numerical method to guide the design of DLD devices, the manuscript is not well organized in its current form, and the simulations were not properly designed. Thus, comprehensive revision is required. Specific concerns are as follows,

1.     The motivation and purpose of this work are not quite clear. There are a lot of existing works investigating the physics of DLD and ways to optimize its performance. Authors need to show how their work is different from others and why their work is important.

2.     Background discussion is not sufficient. Since this work is focused on numerical modeling on DLD, more discussion and summary of similar works should be included in introduction. Authors are also recommended to cite other relevant new works when discuss cell separation/cell sorting in the introduction, for example,

(a)   Zhou et al. Scientific Reportsvolume 8, Article number: 9411 (2018)

(b)   Zhou et al. Microsystems & Nanoengineeringvolume 5, Article number: 8 (2019).

3.     The conclusions are not well supported by their data. (a) Authors claim row shift dictates whether RBCs enter “displacement mode” or “zigzag mode”. However, such conclusion was drawn from the results of particles rather than actual RBCs. Simplified RBC model is not sufficient, especially in 2D model. 3D structure of RBCs and alignment of RBC to flow are not considered.  Why larger than 1.5um not other value in terms of critical row shift? Also, these results are from simulated particles with sphere form rather than the shape of RBCs. The critical row shift could also be different for different designs, e.g. larger pillar size. (b) “Velocity and surface stress” statement also lacks convincing data. It is very difficult to tell if there is significant difference of velocity between circular and triangular DLD devices from Fig.9.  Also, no separation was demonstrated in this work and thus it is misleading to claim “enhance the separation”. (c) Authors did not show solid evidence to support their conclusion of capillary number being used to determine separation mode. Again, no separation was present.

4.     Confusion about displacement mode and zigzag mode.  According to Line 240-241, at low row shift, the device is working in “displacement mode” which is also described in Line 244-245. But the caption of Fig. 4 is contrasting these statements. Please clarify.

5.     Line 272-279, authors used only two row shifts (1.0 and 1.4 μm) to draw the conclusion of the critical row shift was 1.4 μm. This is not rigorously justified as the critical value can be 1.6 or even 2 μm. More results are necessary to determine the critical row shift.

6.     It is not clear if 2D or 3D model was used. Fig. 1 shows 3D structure of DLD devices but all following data/Figures imply 2D model was used. Please clarify.

7.     Deformability of RBCs was ignored and the alignment of RBCs to flow stream was not considered. RBCs can have many different shapes when moving in flow. The challenge of accurately simulating dynamics of RBCs mainly lies on their deformability. As a result, without considering the deformability and flow alignment, the applicability of this work for RBCs in real device can be problematic.  Plus, the work was discussing mainly on particle movement in DLD device rather than RBCs.

8.     As no separation result was present, the title is misleading;

9.     Many unsound/conflicting statements. For example, Line 338, Fig.9 cannot tell if velocity in triangular device was larger.

10.  Fig.7 shows the stress can be up to 800 N/m2, equivalent to 8000 dyne per cm2. This can be detrimental to cell as cells might be damaged when stress is above 100 dyne per cm2. Please refer to the follow paper: Zhou et al. Biomicrofluidics. 2014 Jul; 8(4): 044112

11.  English need be significantly improved. Many times, it is difficult to understand what authors try to convey, for example, line 316-323.

Author Response

Please find the enclosed PDF file.

Reviewer 2 Report

Dear authors,

This manuscript was well prepared and written. The structure of the study was very solid and demonstrated the proposed scheme. It would be better if the authors could compare the simulation data with the experimental results.

Author Response

We appreciate the Reviewer’s suggestions. Due to the complexity of the experiment, most of the existing work is theoretical. To ensure the accuracy of our simulation data, we compare them with the simulation results in the literature. We also hope to compare them from an experimental perspective in the future. 

Round 2

Reviewer 1 Report

The manuscript has been revised. Here are additional comments. (1) Since the simulation is essentially in 2D format, authors are recommended to clarify this in their manuscript. (2) Also, Line 322, is it Figure 3 or Figure 7 cited?